# Flash-Like Albuminuria in Acute Kidney Injury Caused by Puumala Hantavirus Infection

**DOI:** 10.3390/pathogens9080615

**Published:** 2020-07-28

**Authors:** Paula Mantula, Johanna Tietäväinen, Jan Clement, Onni Niemelä, Ilkka Pörsti, Antti Vaheri, Jukka Mustonen, Satu Mäkelä, Tuula Outinen

**Affiliations:** 1Department of Internal Medicine, Tampere University Hospital, 33521 Tampere, Finland; johanna.tietavainen@tuni.fi (J.T.); ilkka.porsti@tuni.fi (I.P.); jukka.mustonen@tuni.fi (J.M.); satu.m.makela@pshp.fi (S.M.); tuula.outinen@gmail.com (T.O.); 2Faculty of Medicine and Health Technology, Tampere University, 33100 Tampere, Finland; onni.niemela@epshp.fi; 3Department of Microbiology, Immunology and Transplantation, Rega Institute, National Reference Center for Hantaviruses, Laboratory of Clinical and Epidemiological Virology, KU Leuven, 3000 Leuven, Belgium; jan.clement.dr@telenet.be; 4Medical Research Unit, Seinäjoki Central Hospital, 60220 Seinäjoki, Finland; 5Department of Virology, Medicum, University of Helsinki, 00100 Helsinki, Finland; antti.vaheri@helsinki.fi

**Keywords:** albuminuria, proteinuria, Puumala virus, hantavirus, acute kidney injury, hemorrhagic fever with renal syndrome, HFRS

## Abstract

Transient proteinuria and acute kidney injury (AKI) are characteristics of Puumala virus (PUUV) infection. Albuminuria peaks around the fifth day and associates with AKI severity. To evaluate albuminuria disappearance rate, we quantified albumin excretion at different time points after the fever onset. The study included 141 consecutive patients hospitalized due to acute PUUV infection in Tampere University Hospital, Finland. Timed overnight albumin excretion (cU-Alb) was measured during the acute phase in 133 patients, once or twice during the convalescent phase within three months in 94 patients, and at six months in 36 patients. During hospitalization, 30% of the patients had moderately increased albuminuria (cU-Alb 20–200 μg/min), while 57% presented with severely increased albuminuria (cU-Alb >200 μg/min). Median cU-Alb was 311 μg/min (range 2.2–6460) ≤7 days after fever onset, 235 μg/min (range 6.8–5479) at 8–13 days and 2.8 μg/min (range 0.5–18.2) at 14–20 days. After that, only one of the measurements showed albuminuria (35.4 μg/min at day 44). At six months, the median cU-Alb was 2.0 μg/min (range 0.6–14.5). Albuminuria makes a flash-like appearance in PUUV infection and returns rapidly to normal levels within 2–3 weeks after fever onset. In the case of AKI, this is a unique phenomenon.

## 1. Introduction

The most common hantavirus infection in Europe is hemorrhagic fever with renal syndrome (HFRS) caused by Puumala virus (PUUV) [1]. The reservoir host of this hantavirus is the bank vole (*Myodes glareolus*), the excreta of which contain the infectious viruses. The virus is transmitted to humans via the inhalation of aerosols contaminated with the virus. Many other hantaviruses, including Dobrava–Belgrade, Sochi and Seoul viruses, are also found in Europe. They cause diseases of varying severity, whereas PUUV predominantly causes a mild form of HFRS. The most severe form of hantavirus infection, the hantavirus cardiopulmonary syndrome (HCPS), occurs in the Americas [2].

After an incubation period of at least two weeks, the symptoms of PUUV infection. Including fever, headache, and back and abdominal pain, occur [3]. The main affected organ is the kidney [4,5]. Soon after the beginning of fever, patients typically present with proteinuria, which reaches the nephrotic range in ~30% of patients [4,6]. Other signs of acute nephritis, including hematuria (in 60–90% of the patients) and rising plasma creatinine, are seen early in the course of the disease, concurrently with thrombocytopenia [4]. Along with predominant acute kidney injury (AKI), clinical signs of capillary leakage, i.e., an acute rise in hematocrit, pleural effusion, ascites, and perirenal fluid accumulation, are often seen [3]. The diagnosis of the disease is based on typical clinical signs and confirmation by serological testing [7].

The care of the disease is supportive, as no specific treatment for PUUV infection exists. The prognosis of the kidney involvement is favorable and recovery from AKI starts when typical polyuria evolves. As a rule, kidney function recovers to normal even after severe AKI, which is seen in ~30% of cases [8]. The need for hemodialysis treatment has been reported to be around 4% [8]. The rare fatal cases (0.1–0.4%) show serious signs of capillary leakage with pulmonary edema and circulatory shock [8]. Severe bleeding complications related to thrombocytopenia are rare [9,10].

While the clinical course of PUUV infection has been well characterized, the pathogenesis of kidney involvement has remained poorly known. Recently, an association was reported between the higher amount of proteinuria and more severe AKI, which raises the question of a possible link between the mechanisms of proteinuria, capillary leakage and AKI in this disease [5,11]. Many mediators of inflammation react in acute PUUV infection. In our earlier studies, the higher urinary excretion of interleukin (IL)-6 and soluble form of urokinase-type plasminogen activator receptor (suPAR), as well as a higher plasma adipocytokine resistin concentration, correlated with the amount of albuminuria [12,13,14].

In our previous study of 70 patients with acute PUUV infection, albuminuria peaked around the fifth day after the beginning of the fever, i.e., 4–5 days before the maximum values of plasma creatinine [6]. There was a clear decline in urine albumin excretion during the hospital stay, but the disappearance of albuminuria was not systemically studied in these patients. Eleven days after the beginning of the fever, three out of six patients still presented with significant albuminuria [6]. Corresponding results were observed in a cohort of 36 patients with acute PUUV infection, in whom a sharp decrease in urine albumin excretion was detected during hospital care [15]. However, at the end of the follow-up, some patients still had albuminuria twelve days after the beginning of the fever [15]. Although earlier reports have shown that long-standing albuminuria is not a typical finding after PUUV-induced AKI [8,16], the timing of the disappearance of albuminuria after acute PUUV infection has not been systemically examined.

The aim of this study was to investigate the disappearance rate of urinary albumin excretion in the course of acute PUUV infection. We examined whether some patients would present with long-lasting albuminuria in order to elucidate the possible underlying factors. We found that regardless of the amount of albuminuria or AKI severity during the acute phase of PUUV infection, albuminuria disappears within 2–3 weeks. Such rapid changes probably reflect the quick alterations in the permeability of the glomerular filtration barrier, an exceptional phenomenon when considering the present knowledge from the other types of AKI.

## 2. Results

The clinical characteristics and laboratory findings of the 141 PUUV-infected patients in the acute phase are shown in Table 1. Out of 141 patients, 44 (31%) had severe AKI, and seven (5%) needed dialysis treatment. Four patients (3%) had symptoms of shock and systolic blood pressure lower than 90 mmHg at admission. One patient had a concurrent acute Guillain–Barré syndrome, probably caused by the PUUV infection, and was treated with plasma exchanges.

Among the 141 consecutive patients, the acute phase urine sample was missing in eight patients due to anuria or technical reasons. Their urine samples were analyzed in the later course of the disease. The acute phase cU-Alb in 133 (94%) patients was measured on the median seventh (range 3–17) day after the beginning of symptoms, i.e., fever. In the acute phase, 40 (30%) patients had cU-Alb 20–200 μg/min. Severely increased albuminuria (cU-Alb 200–1200 μg/min) was detected in 50 (38%) patients and nephrotic range albuminuria (cU-Alb >1200 μg/min) in 26 (20%) patients. Among those who had cU-Alb >200 μg/min, the sample was taken at a median of 7 (range 3–13) days after the onset of fever.

During the hospital stay in the acute phase, 38 patients had only a mild or no clear rise in plasma creatinine concentration, defined by the maximum plasma creatinine under 100 μmol/L. In this group, median cU-Alb was 66.4 (range 2.2–1460) μg/min. Of them, 10 patients had severely increased albuminuria (cU-Alb >200 μg/min) including one subject who had nephrotic range albuminuria (cU-Alb 1460 μg/min).

After hospital discharge, cU-Alb was measured once or twice in 94 patients in the convalescent phase and in 36 patients at six months. The amount of albuminuria at different time points after the beginning of fever is shown in Table 2. At 14–20 days after the beginning of fever, none of the patients had significant albuminuria anymore. The disappearance of albuminuria is shown in the Figure 1.

The amount of albuminuria during hospitalization did not have an effect on the disappearance rate, as shown in Table 3, where the timing and the number of patients at each time point are presented according to the amount of albuminuria in the acute phase. The effect of AKI severity was evaluated in 44 patients with severe AKI, in whom 19 patients’ cU-Alb was controlled within 28 days after the beginning of fever. Those 19 patients had median cU-Alb 337.9 (37.3–6460.7) μg/min at the acute phase, but none of them had significant albuminuria at the first control visit (median cU-Alb 5.5 μg/min, ranging from 1.0 to 18.2 μg/min). Later in the convalescent phase, only one patient had moderately increased albuminuria (cU-Alb 35.4 μg/min, at day 44 after the onset of fever). This patient did not take part in the other follow-up visits and the urine sample was missing in the acute phase due to technical reasons. The maximum plasma creatinine during hospitalization in this patient was 138 μmol/L.

In the acute phase, only three (2%) patients had an U-Alb value lower than the detection limit (3 mg/L). After hospitalization and at six months, 109/147 (74%) and 23/36 (64%) of the U-Alb measurements were under the detection limit, respectively.

## 3. Discussion

The present data show that albuminuria, detected in 89% of the patients in this cohort, disappears rapidly, within two to three weeks after acute PUUV infection. This was observed even in patients with severely increased albuminuria, as well as in patients with severe AKI. To our knowledge, the rate of decline of albuminuria shortly after PUUV infection has not been systematically studied before.

In the acute PUUV infection, the rapid onset of urinary albumin excretion to several grams per day reflects the changes in the barrier function of the glomeruli. A previous analysis of urinary proteins in this disease indicated that the protein is mostly albumin but also larger proteins like IgG are found in the urine, suggesting proteinuria of glomerular origin [17]. The elevated levels of lower molecular weight proteins, α_1_-microglobulin and β_2_-microglobulin, are also found in the urine of PUUV patients reflecting impaired tubular reabsorption capacity [17,18]. In a recent study, also high amounts of free immunoglobulin light chains were detected in the urine during hospitalization in acute PUUV infection [19]. Of note, temporary proteinuria of such a magnitude is not typical for acute tubulointerstitial nephritis (ATIN), which is the most common histological finding in PUUV infection [4].

Endothelial dysfunction and the increased permeability of the capillary wall are characteristic findings during PUUV infection that may affect several organs. One third of the patients have pulmonary infiltrates in chest radiography [20]. Pleural and pericardial fluid and the swelling of the kidneys are signs often found in ultrasound examinations [5,21,22]. Although hantavirus enters the endothelial cells via β_3_ integrin, and endothelial cells in the capillaries of various organs are the main site of hantavirus replication, the virus does not seem to cause direct cytopathic effects [23,24]. This finding corresponds to the strikingly favorable prognosis of kidney function in the course of PUUV infection. Renal biopsy samples in PUUV infection show only minimal glomerular changes, and the main findings include interstitial inflammation, edema and cell-infiltrates, with accompanying acute tubular necrosis to a varying extent (14–88%) [25,26,27]. In a large biopsy material, the histological findings correlated only slightly with the severity of AKI [27]. The precise mechanisms of AKI during acute PUUV infection still remain unresolved.

In other kidney diseases with transient nephrotic range proteinuria, like minimal change disease, a podocytopathy of the glomeruli is often present [28]. A disturbance in the integrity of the glomerular capillary wall during PUUV infection is supported by the results of electron microscopy studies of kidney biopsies taken in the acute phase of the disease [26,29]. Recently, Boehlke et al. reported a patient with acute PUUV infection, biopsied around the seventh day after the onset of fever. The renal biopsy sample showed the effacement of the podocyte foot processes and the loss of slit diaphragms [30]. Furthermore, changes in proteins related to cell-to-cell tight junctions have been reported in the course of acute hantavirus infection [31]. PUUV has been shown to infect podocytes, induce modulations in their cytoskeleton, and impair the migration and adhesion capacity of renal cells in vitro [24,32].

The virulence factors of hantavirus infections and the mediators that are responsible for the varying disease severity remain unknown. Nevertheless, the inflammatory responses of the host are considered important. A multitude of inflammatory mediators are probably activated during the infection, while certain human leukocyte antigen (HLA) -types are associated with a more severe disease during PUUV infection [33]. However, proteinuria of a corresponding magnitude is not typical for many other inflammatory diseases like septic kidney injury. Whether some inflammation factors have a pathogenic role in the emerging proteinuria during PUUV infection remains unclear. Previously, urinary-IL-6 level correlated with the amount of albuminuria in PUUV-infected patients in the acute phase, while in parallel, the plasma level of IL-6 was moderately elevated [12]. The level of urinary suPAR in PUUV-infected patients correlated with the amount of albuminuria, and simultaneously, the plasma concentration of suPAR was also markedly increased [13]. As neither the amount of IL-6 nor suPAR in the urine correlated with their plasma concentrations, these mediators were probably also produced locally in the kidneys. Moreover, high plasma levels of the adipocytokine resistin correlated with the amount of albuminuria, determined by urine dipstick analysis at hospital admission, and also with the severity of the upcoming AKI in PUUV-infected patients [14]. Resistin is known as a marker of macrophage activation and considered as a possible link between AKI and inflammatory responses [34]. The site of excess resistin synthesis during PUUV infection remains unknown. Whether a specific inflammatory mediator could cause albuminuria in PUUV infection is not clear. Of note, nephrotic range proteinuria is a rare finding during other infections and inflammatory diseases accompanied with AKI.

In PUUV infection, the flash-like appearance of albuminuria and the association between albuminuria quantity with upcoming AKI severity [6,11] are unique findings in the context of AKI. In addition, the amount of hematuria associates with the severity of AKI in PUUV infection [35]. The presence of glucosuria, detected in 12% of the patients, was a strong predictive factor not only for clinical shock, but also the severity of AKI in acute PUUV infection [36]. As the current definition of AKI by the committee of Kidney Disease Improving Global Outcomes (KDIGO) leaves urine analysis unacknowledged, it is difficult to compare the PUUV-induced AKI with other forms of AKI [37]. Altogether, AKI comprises a heterogenic group of conditions and the pathogenesis may not be totally consistent. Kidney diseases accompanied with nephrotic range proteinuria are usually related with normal or chronically reduced glomerular filtration rate and a subacute or chronic disease course, and not with AKI. Rare cases of non-steroidal anti-inflammatory drug (NSAID)-induced nephrotic range proteinuria with and without AKI are recognized, but the more usual AKI provoked by NSAID exposure does not present with severely increased proteinuria [38,39]. The PUUV-induced kidney disease also differs from typical ATIN, which is rarely characterized by nephrotic range proteinuria [40]. In PUUV-induced AKI, severely increased, even nephrotic range albuminuria can be present, even with no increase in serum creatinine, as was shown in the present study.

Earlier reports about the disappearance rate of proteinuria after acute HFRS are scarce. Serial measurements of protein excretion were examined in the acute phase of Dobrava–Belgrade virus-caused HFRS in 34 patients, so that three measurements of urinary albumin, IgG and α_1_-microglobulin were performed between hospital admission and 17 days thereafter [41]. The outcome was that both glomerular and tubular proteinuria declined remarkably during hospitalization, compatible with our previous finding in PUUV infection [6], but the timing in relation to the onset of fever or urinary findings after hospitalization were not reported. Cohort studies investigating urinary findings several years after acute PUUV infection show that long-lasting proteinuria is very uncommon, but some patients present with a minor amount of tubular proteinuria (urinary α_1_-microglobulin) [16,42].

Albuminuria in the acute phase of PUUV infection, even when massive, disappears quickly during a period of two to three weeks after the onset of the symptoms. This type of flash-like albuminuria has not been found in other forms of AKI. We conclude that the renal manifestation of this viral infection has unique features compared to other forms of infection-associated AKI and to other proteinuric kidney diseases. Many inflammatory markers are strongly activated during PUUV infection and some of them associate with the amount of albuminuria. Both functional and structural, tubulointerstitial and glomerular changes are documented in this infection. Improved knowledge on the renal pathogenesis of this infection-related AKI could also increase our understanding about the mostly unsolved pathogenesis of other common forms of AKI. There are hardly any previous investigations about the significance of possible transient urinary findings (except the amount of diuresis) related to AKI. Although the severity of AKI in PUUV infection is associated with the amount of albuminuria in the acute phase, neither the amount of albuminuria, nor the severity of AKI, seem to affect the disappearance rate of albuminuria. When investigating the mechanisms of AKI and albuminuria in PUUV infection in the future, the exact timing of the study samples in relation to the onset of fever and albuminuria peak should be taken into consideration.

## 4. Materials and Methods

The study cohort consisted of 141 consecutive patients treated in Tampere University Hospital, Finland, due to acute PUUV infection, between January 2000 and December 2014. These patients also participated in our previous studies [15,43,44,45]. All patients provided written informed consent. The study was approved by the Ethics Committee of Tampere University Hospital (study codes 99256, R04180, R09206), and was conducted in accordance with the Declaration of Helsinki.

The median age of the patients was 42 (range 21–73) years and 96 (68%) were males. Detailed medical history was recorded and physical examination made during hospitalization. The patients had the following diagnoses before acute PUUV infection: hypertension (n = 16), asthma (n = 6), diabetes mellitus type II (n = 5), coronary artery disease (n = 5), rheumatoid arthritis (n = 4), atrial fibrillation (n = 4) inflammatory bowel disease (n = 2), epilepsy (n = 1), celiac disease (n = 1), hypothyroidism (n = 1), sarcoidosis in remission (n = 1), history of melanoma (n = 1), polyneuropathy (n = 1), history of splenectomy (n = 1), sleep apnea (n = 1), multiple sclerosis (n = 1) and transient ischemic attack (n = 1). None of the patients had prior diagnosis of chronic kidney disease.

Acute PUUV infection was serologically confirmed in all patients either from a single serum sample by the detection of the typical granular staining pattern in the immunofluorescence assay (IFA), and/or a low avidity of IgG antibodies to PUUV, and/or PUUV IgM antibodies using an “in-house” enzyme-linked immunosorbent assay based on a recombinant antigen. The diagnostic methods have been previously described [46].

Timed overnight urine samples were collected for the determination of urinary albumin excretion (cU-Alb) once in the acute phase, once or twice in the convalescent phase 10–28 days after hospital discharge, and once at six months after the acute infection. The number of cU-Alb measurements in one patient ranged from one to three (median two).

Urine samples were conserved frozen at −70 °C. The determination of urine albumin (U-Alb) was made by an immunoturbidometric method on a Cobas C 702—Clinical chemistry analyzer (F. Hoffman—La Roche Ltd., Base, Switzerland). The detection limit of the assay for albumin was 3 mg/L. The overnight urinary albumin excretion (cU-Alb) was calculated using the volume of urine collected in a recorded timeframe (minutes) and expressed as μg/min.

Normal physiologic albuminuria was defined as cU-Alb <20 μg/min, moderately increased albuminuria as cU-Alb 20–200 μg/min, and the levels of >200 μg/min were designated as severely increased albuminuria according to the KDIGO guideline [47]. Nephrotic range proteinuria (>3 g/d) was designated as cU-Alb >1200 μg/min.

Plasma creatinine was determined using Cobas Integra (Roche), and severe AKI (stage 3) was defined according to the KDIGO definition by highest plasma creatinine >353.6 μmol/L during hospitalization [37].

Blood cell count was determined using hematological cell counters (Bayer Diagnostics, Elkhart, IN, USA), and sodium, potassium, urea, and albumin concentrations using routine automated chemistry analyzers in the Laboratory Centre of the Pirkanmaa Hospital District (later named Fimlab Laboratories), Tampere, Finland.

The statistical analyses were performed using SPSS (version 20) statistical software (IBM, Chicago, IL, USA). Medians and ranges were given for skewed continuous variables, and numbers and percentages for categorical variables.

## Figures and Tables

**Figure 1 pathogens-09-00615-f001:**
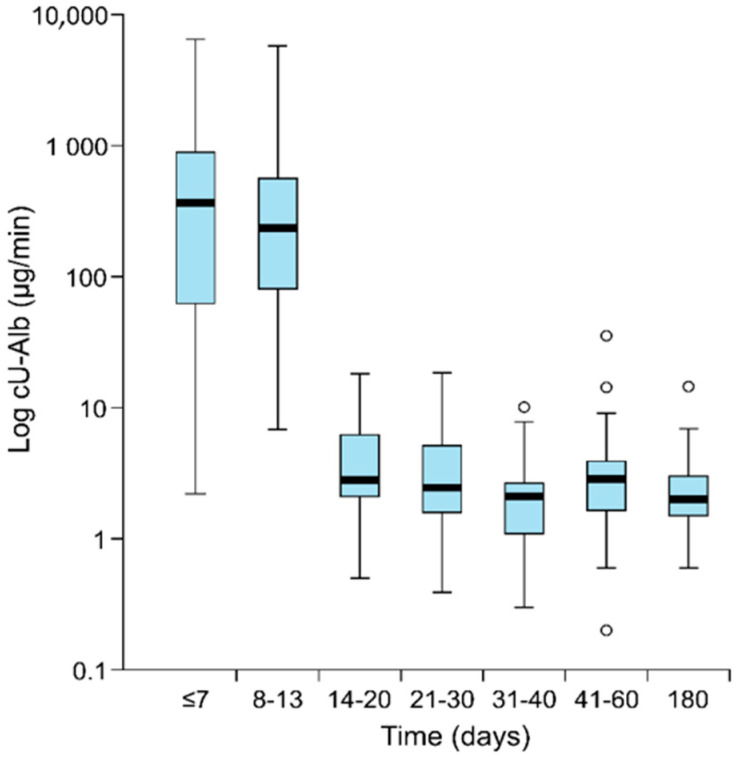
The decrease in overnight albuminuria according to the days after the onset of the fever in 141 patients. Boxplots are with the median (thick line), interquartile range (box), minimum, and the maximum within a 1.5 interquartile range (whiskers). Outliers are displayed as circles. The values of the timely collected overnight excretion of urinary albumin (cU-Alb) are log transformed.

**Table 1 pathogens-09-00615-t001:** Clinical and laboratory findings during hospitalization in 141 patients with acute Puumala hantavirus infection.

Finding	Median	Range
Length of hospital stay (days)	6	2–22
Change in body weight during hospital stay (kg)	2.0	0–12.0
Systolic blood pressure on admission (mmHg)	124	70–210
Plasma creatinine max ^1^ (μmol/L)	185	51–1499
Hematocrit max ^1^	0.44	0.33–0.60
Platelets min ^2^ (×10^9^/L)	61	8–238
Plasma sodium min ^2^ (mmol/L)	130	109–141
Plasma potassium max ^1^ (mmol/L)	4.2	3.3–5.5
Blood leukocytes max ^1^ (×10^9^/L)	10.5	3.9–45.0
Plasma C-reactive protein max ^1^ (mg/L)	79	16–269
Plasma albumin min ^2^ (g/L) n = 45	26	11–36

^1^ max, maximum value during hospitalization; ^2^ min, minimum value during hospitalization.

**Table 2 pathogens-09-00615-t002:** The amount of overnight albuminuria measured at the different time points after the onset of symptoms, i.e., beginning of the fever, in 141 Puumala virus (PUUV)-infected patients.

Days after the Beginning of Fever	≤7 Median 6 Days	8–13 Median 9 Days	14–20 Median 19 Days	21–30 Median 24 Days	31–40 Median 38 Days	41–60 Median 46 Days	6 Months
	n = 77	n = 56	n = 41	n = 32	n = 27	n = 47	n = 36
cU-Alb, μg/min median (range)	311.4 (2.2–6460.7)	234.9 (6.8–5479.2)	2.8 (0.5–18.2)	2.5 (0.4–18.4)	2.1 (0.3–10.1)	2.9 (0.2–35.4)	2.0 (0.6–14.5)

**Table 3 pathogens-09-00615-t003:** The amount of albuminuria at the control visits in 133 PUUV-infected patients categorized by the amount of albuminuria during hospitalization according to the time after the onset of symptoms, i.e., beginning of the fever.

		Days after the Beginning of Fever				
cU-Alb Categories	cU-Alb during Hospitalization, Median (Range)	14–20 Median 19 Days	21–30 Median 24 Days	31–40 Median 38 Days	41–60 Median 45 Days	6 Months
		Number of Patients and cU-Alb at Control Visits, Median (Range)				
	n = 133	n = 41	n = 31	n = 24	n = 44	n = 34
<20 μg/min	n = 17	n = 7	n = 4	n = 4	n = 3	n = 6
	11.4 (2.2–19.0)	3.5 (2.4–9.0)	2.1 (1.9–2.5)	2.0 (1.0–4.6)	1.5 (0.8–1.8)	1.8 (1.4–6.9)
>20–200 μg/min	n = 40	n = 19	n = 5	n = 10	n = 15	n = 9
	86.4 (21.4–198.4)	2.2 (0.5–14.6)	1.8 (0.4–8.0)	2.4 (0.7–6.3)	3.2 (0.2–5.4)	1.5 (1.1–5.4)
>200–1200 μg/min	n = 50	n = 8	n = 14	n = 7	n = 19	n = 11
	410.1 (203.1–974.8)	4.6 (0.8–14.5)	2.4 (0.5–18.4)	1.8 (0.5–7.8)	2.6 (1.0–7.8)	2.6 (0.6–14.5)
>1200 μg/min	n = 26	n = 7	n = 8	n = 3	n = 7	n = 8
	1981.1 (1202.5–6460.7)	3.8 (1.3–18.2)	4.9 (1.0–8.0)	2.5 (2.1–10.1)	3.3 (1.7–14.3)	2.0 (1.3–3.1)

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
