# Peer review of "Flash-Like Albuminuria in Acute Kidney Injury Caused by Puumala Hantavirus Infection"

_pathogens, 2020, doi:10.3390/pathogens9080615_

Round 1
Reviewer 1 Report
The authors correctly state that "improved knowledge of the renal pathogenesis of this infection-related AKI, could also increase our understanding". This is certainly true. However, the authors also state "When investigating the mechanisms of AKI and albuminuria in PUUV in the future....." To make a useful paper some hypothesis should be tested or mechanism described and not left till later. Purely descriptive papers, which overlap to a certain extent with prior publications, and where only one or two parameters are described are simply neither particularly interesting nor valuable.
Author Response
Please see the attachement.

Reviewer 2 Report
The authors responded to all comments.
Now, the manuscript is well written and organized.
Author Response
Please see the attachement.

Reviewer 3 Report
The manuscript by Mantula et al. describes the kinetics of albuminuria in a large cohort of Puumala virus patients. The manuscript is well-written and expands on previous reports on this topic, with the main advance being a larger number of patients sampled at multiple time points.
Minor concerns: The manuscript would be improved with a table or figure breaking down the cU-Alb over time in sub-groups of patients (those with <20 ug/min, 20-200 ug/min, 200-1200 ug/min, and >1200 ug/min).
Author Response
Please see the attachement.

This manuscript is a resubmission of an earlier submission. The following is a list of the peer review reports and author responses from that submission.
Round 1
Reviewer 1 Report
The purpose of this study was to investigate the disappearance rate of urinary albumin excretion in the course of acute PUUV infection. The authors found that regardless of the amount of albuminuria or AKI severity during the acute phase of PUUV infection, albuminuria disappears within 2-3 weeks.
Comments:
- Determine the dispersion parameters of your study such as the standard deviation.
- Represent graphically the data of your study.
- Compare your results obtained in this paper with other existing in the literature.
- Check the references of your study.
Reviewer 2 Report
The authors describe the time course of proteinuria in patients suffering from PUUV HFRS. The study is of sufficient size to have predictive power. The data is correctly analysed and the conclusions appear firm. The paper is also well-written. The term cU-Alb is not clearly defined, although it is self-explanatory.
The data is in some regards similar to the data the same first author published in 2017 Nephron, in which a study group of 205 patients were examined in the acute phase of infection for overnight urinary protein and albumin release. This has been noted by the authors, however. The principle point of this paper is that the rate of decrease and normalisation in cU-Alb in the recovery phase has not as yet been described. While this is novel it is of limited interest as it is already known that this parameter normalises after infection, again as noted by the authors. What significance the rate of normalisation has is not defined in this paper, nor is the mechanism investigated.
In summary, this paper points out that this disease has novel, unique features. However, this paper does not contribute to the understanding or significance of these features, and is rather a descriptive paper.